# Giant photothermal nonlinearity in a single silicon nanostructure

Yi-Shiou Duh[1,11], Yusuke Nagasaki[2,11], Yu-Lung Tang[1], Pang-Han Wu[1], Hao-Yu Cheng[3], Te-Hsin Yen[1], Hou-Xian Ding[1], Kentaro Nishida[2,4], Ikuto Hotta[2], Jhen-Hong Yang [5], Yu-Ping Lo[6], Kuo-Ping Chen[6], Katsumasa Fujita[2,4], Chih-Wei Chang [7], Kung-Hsuan Lin [3✉], Junichi Takahara [2,8✉] & Shi-Wei Chu [1,9,10✉]

Silicon photonics have attracted significant interest because of their potential in integrated photonics components and all-dielectric meta-optics elements. One major challenge is to achieve active control via strong photon–photon interactions, i.e. optical nonlinearity, which is intrinsically weak in silicon. To boost the nonlinear response, practical applications rely on resonant structures such as microring resonators or photonic crystals. Nevertheless, their typical footprints are larger than 10 μm. Here, we show that 100 nm silicon nano-resonators exhibit a giant photothermal nonlinearity, yielding 90% reversible and repeatable modulation from linear scattering response at low excitation intensities. The equivalent nonlinear index is five-orders larger compared with bulk, based on Mie resonance enhanced absorption and high-efficiency heating in thermally isolated nanostructures. Furthermore, the nanoscale thermal relaxation time reaches nanosecond. This large and fast nonlinearity leads to potential applications for GHz all-optical control at the nanoscale and super-resolution imaging of silicon.

[1] Department of Physics, National Taiwan University, 1, Sec 4, Roosevelt Rd., 10617 Taipei, Taiwan. [2] Graduate School of Engineering, Osaka University, 2-1 Yamadaoka, Suita, Osaka 565-0871, Japan. [3] Institute of Physics, Academia Sinica, 128, Sec. 2, Academia Rd., 11529 Taipei, Taiwan. [4] AIST-Osaka University Advanced Photonics and Biosensing Open Innovation Laboratory, AIST, 2-1 Yamadaoka, Suita, Osaka 565-0871, Japan. [5] Institute of Photonic System, National Chiao Tung University, 301 Gaofa 3rd Road, 711 Tainan, Taiwan. [6] Institute of Imaging and Biomedical Photonics, National Chiao Tung University, 301 Gaofa 3rd Road, Tainan 711, Taiwan. [7] Center for Condensed Matter Sciences, National Taiwan University, 1, Sec 4, Roosevelt Rd., 10617 Taipei, Taiwan. [8] Photonics Center, Graduate School of Engineering, Osaka University, 2-1 Yamadaoka, Suita, Osaka 565-0871, Japan. [9] Molecular Imaging Center, National Taiwan University, 1, Sec 4, Roosevelt Rd., 10617 Taipei, Taiwan. [10] Brain Research Center, National Tsing Hua University, 101, Sec 2, Guangfu Road, 30013 Hsinchu, Taiwan. [11] These authors contributed equally: Yi-Shiou Duh, Yusuke Nagasaki. ✉email: linkh@sinica.edu.tw; takahara@ap.eng.osaka-u.ac.jp; swchu@phys.ntu.edu.tw

Because of its natural abundance and its compatibility with industrial production lines, silicon is the most widely used material in the modern electronics industry. However, because of its indirect bandgap, Si has limited applications in photonics. It is a long-awaited goal to combine photonics with the advantages of silicon. It was not until the recent decade that we witnessed dramatic progression in silicon photonics, including amplification, lasing, and super-continuum generation[1–3].

In the field of silicon photonics, one particular emphasis is placed on achieving all-optical control, which requires strong photon–photon interactions or optical nonlinearity[3,4]. Conventionally, Kerr-type nonlinearities provided ultrafast response. However, the magnitude of nonlinearity ($n_2$) is on the order of $10^{-9}$ $\mu m^2/mW$[3]. Photothermal effects are known to provide much larger nonlinear responses, with the effective nonlinear coefficient $n_2$ approaching $10^{-6}$ $\mu m^2/mW$[5]. From a simple estimation based on $n = n_0 + n_2 I$, in which n is the overall refractive index, $n_0$ is the linear index, $n_2$ is the nonlinear index, and I is excitation intensity, to create 10% nonlinear deviation from the linear response, an exceedingly high excitation intensity in the order of $10^5$ $mW/\mu m^2$ is necessary.

Because of the weak nonlinearity of Si, the typical design of nonlinear silicon photonic components requires resonant structures such as microring resonators and photonic crystals[6,7]. However, the cost of high quality factor (Q) resonant structures is their large feature size, usually on the order of 10 μm. For example, on the recently published silicon electronics and photonics dual platform[8], the photonic microring resonator was much larger than the electronic transistors.

Inspired by metal-based plasmonics, an emerging field is focused on significantly enhancing the light-matter interactions via strong light confinements using nanoscale high-index dielectrics[9], without being compromised by metal loss. The meta-silicon-material has led to various unexpected optical properties, e.g., Mie-resonance-induced localization as well as electric/magnetic dipole/multipoles[4], optical magnetism[10], directional emission[11], broadband perfect reflector[12], high-efficiency hologram[13], optical topological states[14], and multicolor nano-display[15]. Nanostructured Si also displayed many unusual optical control[16,17] or nonlinearity mechanisms[18]. For example, Si metasurfaces can be applied for fabrication of ultracompact phase controllers[17], several order-of-magnitude enhancement in third harmonic generation[19,20], and two-photon absorption[21]. However, the optical nonlinearities are still far from sufficient to realize applications such as all-optical control at a low light intensity.

In this study, we combine Mie resonance with the photothermal effect and report the unexpectedly large photothermal nonlinearity in a single Si nanostructure of ~0.001 μm³ volume. The single-nanostructure nonlinearity enables a 400% enhancement or 70% reduction in scattering, i.e., a significant deviation from the linear response, at an excitation intensity of merely 1–10 $\mu m^2\,mW^{-1}$. The equivalent $n_2$ reaches 0.1 $\mu m^2\,mW^{-1}$, which is five orders of magnitude larger than the photothermal nonlinearity of bulk silicon, providing 90% reversible and repeatable all-optical modulation in a single silicon nanostructure. The thermal relaxation time of the nanostructure-based nonlinear response is on the order of nanoseconds, leading to the potential of GHz operation. The large nonlinearity and the fast response are promising for all-optical nano-silicon applications. Furthermore, we demonstrate significant point spread function reduction via the giant nonlinearity, that holds great potential toward label-free super-resolution imaging of silicon.

## Results

### Size-dependent optical properties of silicon nanostructures.
The sample was a single-crystalline Si nanoblock array on quartz (see Fig. 1a and fabrication detail in Methods), which has been demonstrated recently to host multipolar electric/magnetic resonances within a single unit[15,22]. Figure 1b is a scanning ion-beam microscope image of one Si nanoblock, showing the high-quality sharp edges and corners. In the array, the nanoblock height was fixed at 150 nm, and the lateral dimensions ($w_x$ and $w_y$) were increased from 60 to 220 nm in 10-nm steps to induce a controllable wavelength shift of Mie resonance, as shown in the colored Fig. 1c (see setup in Methods and Supplementary Fig. 1). The distance between each nanoblock is 5 μm, and thus, the coupling among them is negligible. Based on the transparent quartz substrate and a 561-nm dark field laser-scanning microscope, Fig. 1d shows the size-dependent scattering intensity along the diagonal nanoblocks ($w_x = w_y$). The choice of this wavelength allows induction of multipole Mie resonances, as shown by the white lines in the size-dependent spectra of Fig. 1e, f (see Supplementary Fig. 2 for multipole decomposition analysis). Figure 1g is the simulated size-dependent scattering at 561 nm, which agrees well with Fig. 1d. Additional theory-experiment correspondences on the single-nanoblock spectrum are shown in Supplementary Fig. 3. It is interesting to notice that by varying the nanoblock size, more than 80% of scattering intensity variation was observed. A more interesting question would be whether similar variations could be found by optically tuning silicon's refractive index, not size, to reach the unprecedented optical nonlinearity.

### Giant nonlinearity of scattering in silicon nanostructures.
The nonlinearity of the Si nanoblock scattering was studied using a laser-scanning microscope (xy-scan, see supplementary Methods) at 561-nm, and the results are shown in Fig. 2. The concept of the xy-scan to characterize nanostructure nonlinearity is adapted from the z-scan. In the z-scan, a sample should be much thinner than the axial (z) length of focus, and when a focused beam scans across the thin sample, deviation from a linear trend indicates the existence of nonlinearity. In xy-scan, the size of nanoblocks has to be much smaller than the lateral (xy) point spread function (PSF), and when a focused laser beam scans laterally across a nanoblock, deviation from a Gaussian profile indicates the nonlinear response[23], as shown by the insets of Fig. 2a–d.

In Fig. 2a, which corresponds to the $w = 100$ nm nanoblock (magnetic dipole dominates, the first peak in Fig. 1d), at low intensity up to 1.5 mW $\mu m^{-2}$, the scattering response is linear. Nevertheless, as the laser intensity increases, scattering starts to saturate, i.e., negatively deviates from the linear trend (red line). We define the nonlinear deviation ratio (NDR) as $\Delta S/S_e$, where $\Delta S$ is the percentage deviation of measured scattering, and $S_e$ is the extrapolated linear response. Accordingly, an NDR of $-50\%$ is obtained at 6 mW $\mu m^{-2}$, whose PSF significantly deviates from the original Gaussian profile (see inset of Fig. 2d, which also shows the $-50\%$ NDR).

Figure 2b presents the scattering nonlinearity of the $w = 170$ nm nanoblock, whose scattering is relatively weak and off-resonance in Fig. 1d. Similar to Fig. 2a, scattering is linear at the low excitation intensity but is considerably different at high intensity; the scattering signal of this weak-scattering particle exhibits a sharp reverse saturation, i.e., increases with a very large slope and then saturates. At 5 mW $\mu m^{-2}$, more than 400% positive NDR is observed.

Figure 2c shows the case of the $w = 190$ nm nanoblock, which exhibits the largest scattering intensity in Fig. 1d. In addition, scattering is linear at low intensity, and at high intensity, the

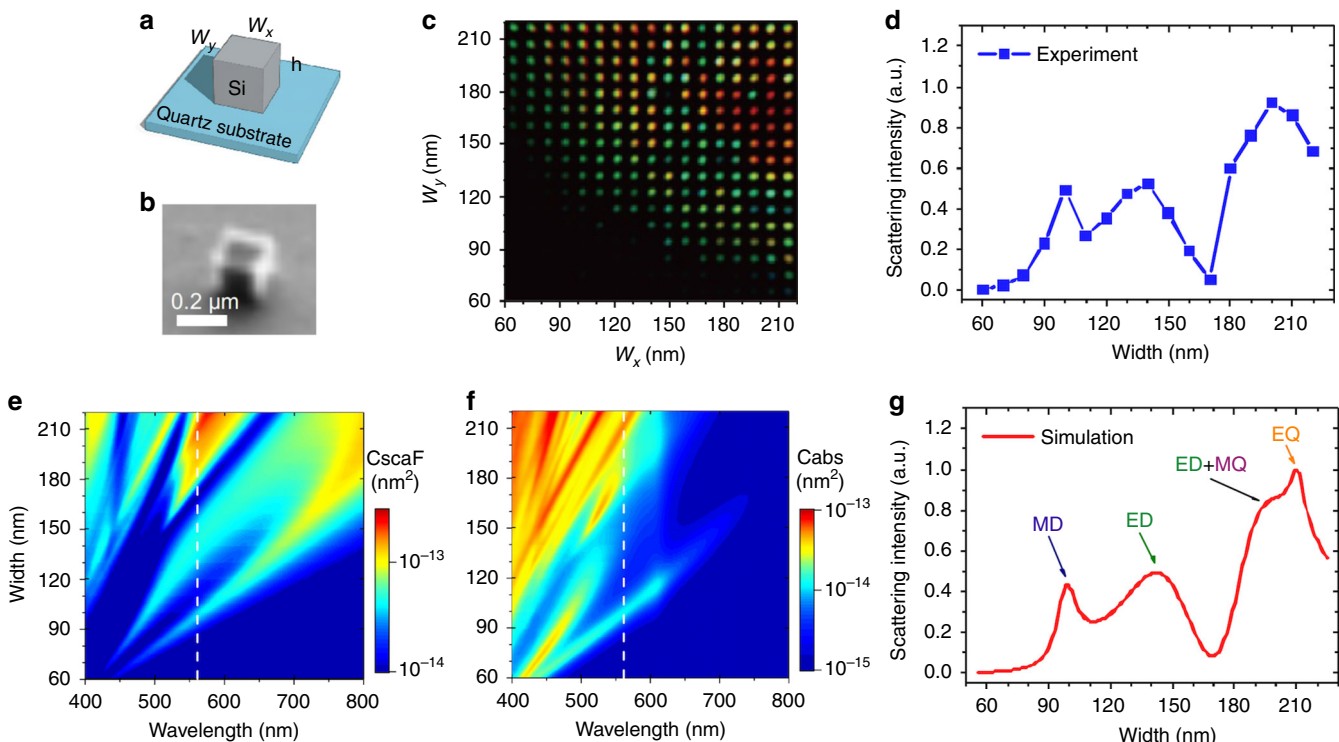

**Fig. 1 Optical properties of Si nanostructures. a** We used Si nanoblocks on a quartz substrate, whose lateral dimensions ($w_x$ and $w_y$) vary from 60 to 220 nm, with a 10-nm step, and a fixed height (h) of 150 nm. **b** Focused ion-beam image of a typical Si nanoblock ($w_x = w_y = 200$ nm). Resolution is compromised because of the charging effect of the non-conductive quartz substrate. **c** Halogen lamp illuminated dark-field image of the isolated Si nanoblocks, showing colorful resonance among different sizes. **d** 561-nm scattering intensity along the array diagonal ($w = w_x = w_y$), presenting the characteristic multipole resonance of silicon nanostructures. The excitation intensity is 1.3 mW μm$^{-2}$. **e** Simulated single-particle forward scattering (CscaF) and **f** absorption cross section (Cabs) spectra versus different nanoblock widths. The white dashed line marks the 561-nm laser, which excites multiple resonances. **g** Simulation of size-dependent 561-nm scattering, which agrees well with experiment results. MD magnetic dipole, ED electric dipole, EQ electric quadrupole, MQ magnetic quadrupole.

scattering response exhibits a negative deviation, similar to Fig. 2a. Nevertheless, here, scattering is "super-saturated", i.e., reduced with increasing excitation, and the minimum NDR reaches −70%.

Apparently, each particle shows different nonlinear responses, and the NDRs at 6 mW μm$^{-2}$ are summarized in Fig. 2d, with corresponding PSF profiles. The reversibility and repeatability of the nonlinear responses are demonstrated in Fig. 2e and Supplementary Fig. 6, thus excluding the possibility of non-reversible oxidation or shape changes of Si[24]. The size-dependent NDR of the whole array is given in Fig. 2f, leading to a few insights.

First, these large nonlinear deviations at relatively low excitation intensities lead to an effective nonlinear index $n_2$ of 10$^{-1}$ μm$^2$ mW$^{-1}$, (see Methods and Supplementary Fig. 8 for derivation). This value is much larger than the reported photothermal nonlinearity of Si ($n_2 \sim 10^{-6}$ μm$^2$ mW$^{-1}$)[5], featuring a five-order improvement with an ultrasmall mode volume of 0.001 μm$^3$.

Second, when Fig. 2d is compared with Fig. 1d, the positive and negative NDR values in Fig. 2d correspond well to the valley and peaks in Fig. 1d, but not vice versa. For instance, the $w = 140$ nm Si nanoblock is the second resonance peak in Fig. 1d, but its NDR is less than 5%, much smaller than that of the other two peaks. The $w = 110$ nm Si nanoblock exhibits a scattering valley in Fig. 1d, but no nonlinearity is observed.

Third, not every nanoblock exhibits NDR; for example, the $w = 120$ nm nanoblock shows a linear response throughout our excitation intensity range (Supplementary Fig. 4). Below, we unravel the mechanism of this huge and anomalous nonlinearity.

**Photothermal mechanism of the giant nonlinearity.** Silicon is known to exhibit various optical nonlinearities, including parametric processes such as frequency mixing and optical Kerr effect, as well as nonparametric processes such as multiphoton absorption, inelastic scattering (e.g., Raman), free-carrier absorption (FCA), and photothermal effect (PT). The nonlinearity magnitudes of the first four are all in the order of 10$^{-8}$–10$^{-9}$ μm$^2$ mW$^{-1}$[25,26], and the last two are known to be the most effective to modify silicon's index. Although FCA in Si can produce an index difference of as large as 0.1[27], it requires strong pulsed excitation. Under our continuous-wave excitation, the free-carrier density is estimated to be $4 \times 10^{13}$ cm$^{-3}$, and the corresponding $n_2$ value is only 10$^{-7}$ μm$^2$ mW$^{-1}$[25], which is much smaller than the value we observed experimentally. Therefore, the photothermal mechanisms should be the dominating mechanism. Recently, Mie-resonance-enhanced photothermal effect was reported to provide a large third-order nonlinearity in sub-100-nm metallic nanostructures[23,28].

The temperature rise during laser excitation is verified through Raman spectroscopic measurement. Figure 3a unravels a few-hundred Kelvin temperature elevation from a single silicon nanostructure (see Methods for derivation), confirming the existence of the photothermal effect. We have characterized the temperature-dependent complex refractive index of silicon by ellipsometry (see Supplementary Fig. 9). In the following, a

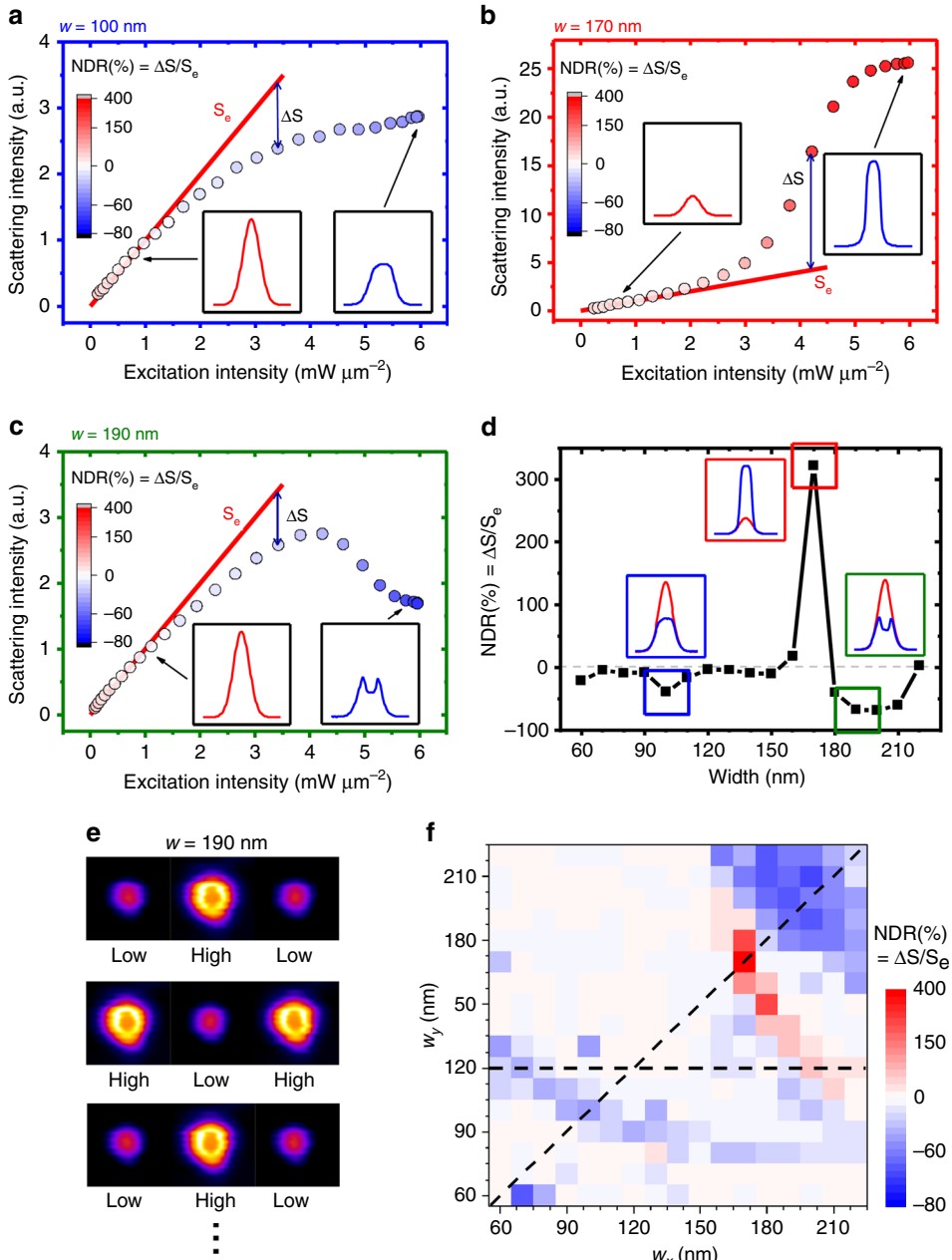

**Fig. 2 Experimental observation of nonlinear scattering. a–c** Excitation intensity-dependent scattering for $w = 100$ nm, 170 nm, 190 nm, observed using a dark-field laser-scanning microscope at $\lambda = 561$ nm. In the main frames, red lines and colored dots indicate linear scattering intensity ($S_e$, extrapolated from low-intensity excitation, see Supplementary Figs. 4 and 5) and measured scattering intensity (whose deviation from $S_e$ is $\Delta S$), respectively. An unexpectedly large nonlinearity is manifested through the significant deviation from linear intensity dependence. The color of the dots represents nonlinear deviation ratio (NDR), i.e., percentage of $\Delta S/S_e$. The insets correspond to PSFs at low and high intensities (lateral distance: 4 μm). **d** NDR versus different sizes of nanoblocks at a 6 mW μm$^{-2}$ excitation intensity. The dotted line marks 100%, which means no nonlinear response. The blue, red, and green rectangles highlight the regions of large NDR. The insets present the corresponding PSF profiles at low excitation (red curves) and high excitation (blue curves) intensities, where the large nonlinearity is again manifested by the large deviation of blue profiles from Gaussian distribution. **e** The PSF recovery during repetitive switching between low-intensity (1.3 mW μm$^{-2}$) and high-intensity (6 mW μm$^{-2}$) excitations, demonstrating reversible and repeatable nonlinear responses (see Supplementary Fig. 6 for other nanoblocks). **f** Experimental NDR map of the whole array. The diagonal dashed line marks the nanoblocks for analysis in Fig. 1d and Fig. 2. The horizontal dashed line marks non-diagonal nanoblocks that are compared to simulation in Supplementary Fig. 7.

detailed simulation based on the photothermal response and scattering of a Mie-resonant silicon nanoblock is carried out, showing outstanding agreement with experiments.

The size- and temperature-dependent absorption cross sections are given in Supplementary Fig. 10, presenting the need for iterative calculation to derive the correct temperature elevation under the photothermal effect (see Methods). The iterative result is given in Fig. 3b, where a few-hundred-Kelvin temperature increase is found, not only indicating that silicon nanostructures are indeed efficient heaters but also agreeing well with our Raman experiment results. The absorption-induced temperature increase in turn affects the refractive index as well as the scattering cross

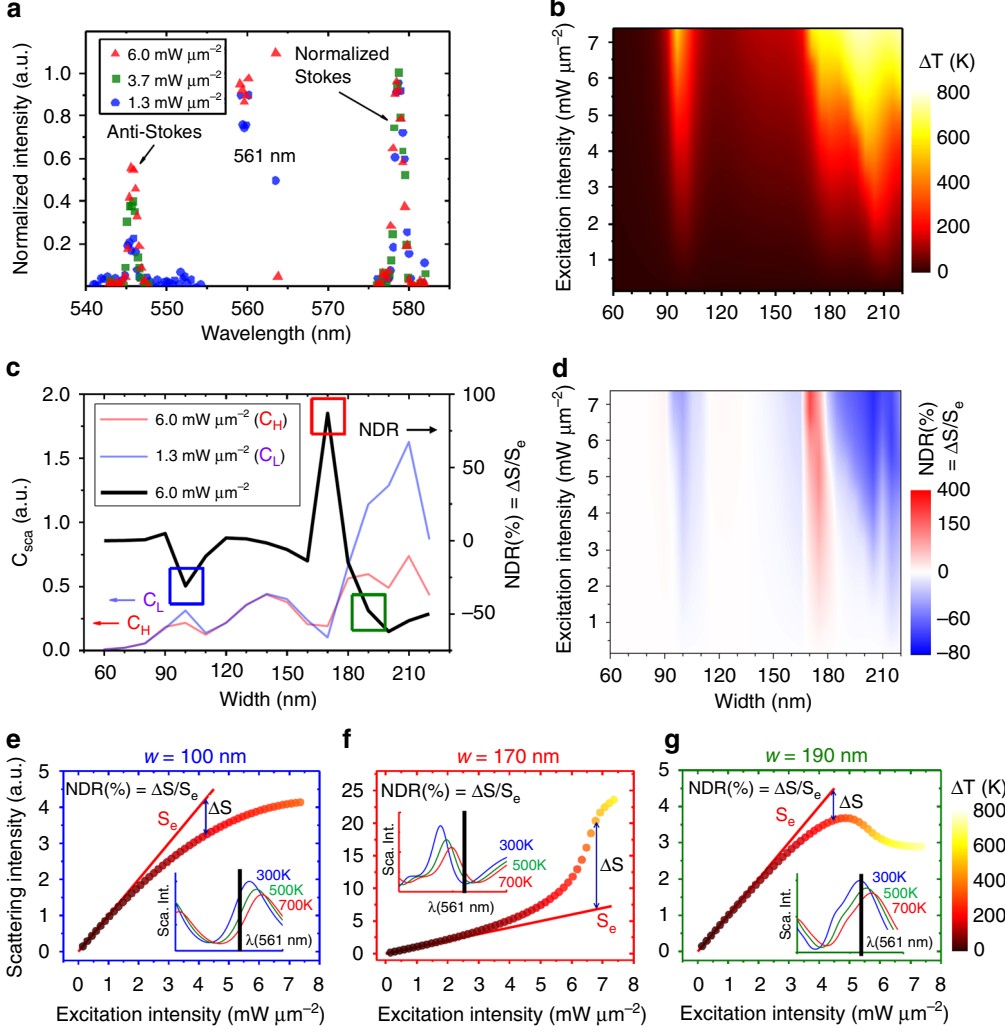

**Fig. 3 Photothermal-based nonlinearity. a** The temperature of an individual nanoblock (w = 190 nm) can be measured in-situ by Raman scattering. With increasing excitation intensity, the ratio of anti-Stokes versus Stokes intensity gradually increases, indicating a temperature increase up to a few-hundred Kelvin (see Methods). **b** Simulated temperature map of Si nanoblocks versus excitation intensity and width, indicating non-uniform heating up to ~1000 K. **c** Simulated 561-nm scattering cross sections at two excitation intensities (light blue $C_L$: 1.3 mW μm$^{-2}$; light red $C_H$: 6.0 mW μm$^{-2}$), and the corresponding NDR (black line, derived from $C_H$ over $C_L$). The size-dependent NDR agrees very well with the experimental results in Fig. 2d. **d** Evolution of NDR versus excitation intensity and nanoblock size. **e–g** Laser intensity-dependent scattering for w = 100, 170, and 190 nm nanoblocks, respectively, again agreeing very well with Fig. 2. The color of each dot indicates the equilibrium temperature under photothermal heating. The inset of each figure presents the corresponding resonance spectrum shift at elevated temperatures.

section, known as the thermo-optic effect, thus leading to the giant nonlinear optical behaviors.

Figure 3c depicts the 561-nm scattering cross sections at low-intensity (1.3 mW μm$^{-2}$, light blue curve) and high-intensity (6.1 mW μm$^{-2}$, light red curve) excitations, manifesting dramatic variations. Their ratio represents the simulated size-dependent NDR, i.e., the black curve in Fig. 3c, which agrees well with experimental results in Fig. 2d. It is understandable now why the 110-nm and 140-nm nanoblocks exhibit diminishing nonlinearity because their heating is not significant. The full intensity-dependent evolution of NDR is given in Fig. 3d, showing large NDR indeed corresponds to large temperature elevation.

Further verification with experimental results is provided in Fig. 3e–g, which are the nonlinear scattering of 100-, 170-, and 190-nm nanoblocks versus excitation intensity. Striking similarities to Fig. 2a–c are found, justifying the correctness of both experiments and the simulations. The negative nonlinear deviation (100-nm, 190-nm) and positive nonlinear deviation (170-nm) are well explained by the temperature-dependent

resonance spectrum shift in the insets. Therefore, we concluded that Mie-resonance-enhanced photothermal effect[28] is the dominating mechanism of the unexpectedly large nonlinear response in the single silicon nanostructure. Thus, it would be an interesting theoretical challenge to find an analytical expression of photothermal nonlinearity versus Q-factor.

One feature of the photothermal effect is the sensitivity to the surrounding. Supplementary Fig. 11 shows the nonlinear response with silicon nanoblocks immersed in glass-index-matching oil, whose thermal conductivity is one order larger than air. Very steep nonlinear variation is found in the intensity dependency, potentially enabling high-contrast all-optical control with a small power variation. More studies will be required to investigate the best shape/size/environment for heating nanoparticles and for inducing the maximal photothermal nonlinearity.

**Applications of photothermal nonlinear scattering.** The giant nonlinearity of a single silicon nanostructure that we report here can be applied to various photonic applications, such as

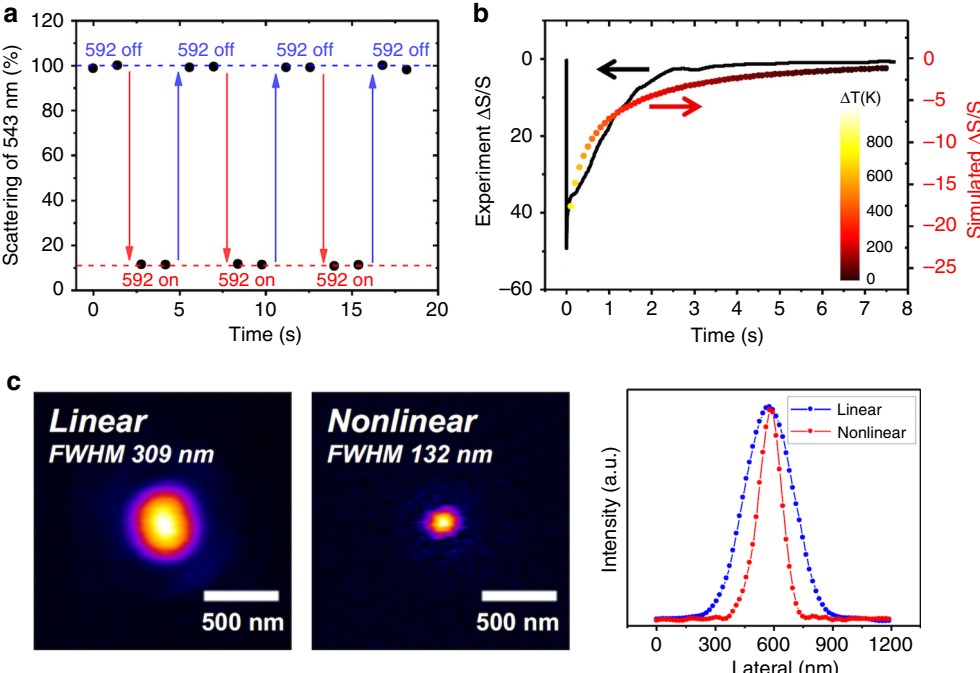

**Fig. 4 Potential applications of photothermal nonlinear scattering. a** All-optical switch on a single nanoblock with simultaneous illumination by a pump (592-nm) and a probe (543-nm) beams. When the pump is on, the probe scattering is efficiently turned off. **b** Temporal evolution of scattering and temperature relaxation in a single nanoblock. The black line is the transient pump-probe experimental result, agreeing well with the simulation (colored line; colors represent temperature). Both show a relaxation lifetime of ~1 ns. **c** Resolution enhancement of a silicon nanostructure via nonlinearity. The size of PSF is reduced by more than 2-fold with nonlinear response. Corresponding cross-sectional line profiles are provided in the right panel.

ultrasmall all-optical switch and super-resolution imaging on silicon, which is shown in Fig. 4. Figure 4a demonstrates the all-optical switch, where the scattering of a probe beam (at 543 nm) from a single silicon nanostructure ($240 \times 240$ nm) can be efficiently switched off using overlapped excitation of a pump beam (at 592 nm). The modulation depth reaches 90%, and it is fully reversible/repeatable.

One important factor in the all-optical switch is speed. Figure 4b shows the transient response of nonlinear scattering deviation via the pump-probe technique (black line, see Methods), and simulation (colored line). The temperature variation of the nanoblock during relaxation is depicted using different colors. Here, we show theoretically and experimentally, that photothermal relaxation time of an isolated silicon nanostructure reaches nanoseconds, i.e., GHz operation potential, with very large modulation depth. We envision to integrate the reversible photothermal nonlinearity into the field of meta-optics to achieve highly desirable all-optical tunability with a thermally isolated silicon nanostructure. Note that for thermally coupled nanostructures on a metasurface, to avoid heat accumulation, their repetition rate has to be reduced to kHz[19].

Figure 4c shows a potential application of the all-optical switch, i.e., super-resolution imaging. It is well known that the capability to precisely control light emission on/off leads to significant resolution enhancement[29], as we have previously shown for plasmonic nanostructures[28]. In contrast, the nonlinear response itself also enables significant resolution enhancement (see supplementary methods)[30]. Here, we demonstrate ×2.3 resolution enhancement in Fig. 4c, i.e., beyond the diffraction limit, with a 100-nm nanoblock. The nanostructure exhibits a strong nonlinear response (saturation of scattering in Fig. 2a). With a Gaussian focus, the nonlinear response should start from the center of PSF, and thus, by extracting the nonlinear part, the resulting PSF becomes smaller than its linear counterpart. This

demonstration extends the application of super-resolution microscopy not only to label-free silicon nanostructure observations but also further into biomedical applications with silicon nanoparticles.

## Discussion

In this work, we discovered a large and fast photothermal nonlinearity of Si nanostructures, enabled by Mie-resonance enhanced absorption and thermally isolated efficient heating. The nonlinear coefficient ($n_2$) is five orders larger over the bulk silicon photothermal nonlinearity and is much larger compared to all previous nonlinear silicon reports, thus allowing more than +400% to −90% nonlinear deviation of scattering under continuous-wave (CW) illumination. Transient measurements and simulations revealed nanosecond thermal dissipation time, without sacrificing the large NDR. Our results open up a new direction in nonlinear silicon nanophotonics for application in high-speed, high-contrast all-optical switches in nanoscale as well as super-resolution imaging of silicon.

The large photothermal nonlinearity is based on efficient heating of silicon nanostructures, which has recently been found as highly effective optical heaters[31], especially when the excitation is resonant with MQ. From Fig. 2 and Supplementary Fig. 3, nanoblocks with excitation at MQ resonance (the 190-nm-width one, see Supplementary Fig. 3b) or excitation at a spectral valley next to MQ resonance (the 170-nm-width one, Supplementary Fig. 3a) demonstrate better nonlinearity. Notably, the Q-factor of MQ resonance is only ~20–30, but the photothermal nonlinearity of the silicon nanoblock is five orders of magnitude larger than that of bulk. This is because Mie resonance enhances nano-silicon absorption by Q = 20–30 times over the bulk, but the temperature rise can be much more than Q times because nano-silicon is surrounded by low-thermal-conductance materials (air and quartz). That is, the greatly enhanced photothermal nonlinearity

is attributed to not only enhanced absorption from Mie resonance, but also to high-efficiency heating due to the thermally isolated environment. Note that our reported photothermal nonlinear response in an isolated low-Q silicon resonator is much higher than that in a thermally coupled nanostructure on a metasurface with free-carrier effect[32] or with Fano resonance (Q-factor ~1000)[19]. The latter provides maximally 10–30% modulation of transmission, whereas we achieved much larger modulation at a few mW μm$^{-2}$.

Conventionally, photothermal effect is considered as a slow response. In Fig. 4b we demonstrated that the photothermal nonlinearity of a single silicon nanostructure can be as fast as nanosecond. Looking into the experimental curve of Fig. 4b, there are two relaxation processes. First, $\Delta S/S$ reaches −50% within the pulse duration (~1 ps) and subsequently relaxes to −35% within 0.1 ns. Since the excited density of free carriers is high ($10^{20}$–$10^{21}$ cm$^{-3}$ under our experimental conditions) and close to the damage threshold, most energy of free carriers is efficiently transferred to the lattice temperature through the Shockley-Read-Hall process and Auger recombination in this duration[33–35]. Following, $\Delta S/S$ is dominated by lattice temperature and reveals a second relaxation which takes a few nanoseconds to zero. This is primarily attributed to thermal dissipation from the Si nanoblock to the surrounding medium. Note that in Figs. 2 and 3, where CW lasers are applied, the laser intensity is 4–5 orders lower than that of Fig. 4b, and thus, the free-carrier effect is negligible. More studies are required to understand the slight differences between simulation and the experiment in the slow process. Furthermore, the optimized responses are expected if more factors such as particle geometries, immersion materials, or phase change effects are considered.

There are several possibilities to increase the modulation speed. One is wavelength division multiplexing, which allows simultaneous signal processing at different wavelengths (Fig. 4a) to enhance bandwidth. In contrast, it is known that at a scale less than 100 nm, thermal conductance is dictated by the ballistic condition, i.e., thermal conductance per unit area becomes constant[36]. Therefore, thermal relaxation time would be directly proportional to height. In our current experiment, a 150-nm height was adopted. Thus, by reducing the height, significant enhancement of speed can be expected. Moreover, the speed may be further enhanced by replacing the substrate with a material of higher thermal conductivity. These will be the future directions of our research.

In addition to the above applications, recently, there has been an emerging trend using thermo-optic effect to modulate metamaterial optical properties, such as polarization[37], miniaturized image contrast[38], and metalens focal length[39]. We envision that our result manifests the potential of pushing these various modulations toward all-optical control.

## Methods

**Sample preparation.** The 150-nm-thick monocrystalline silicon on quartz substrate (Shin-Etsu Chemical Co., Ltd.) was fabricated by wafer bonding at the temperature of T < 1000 degree after H + ion implantation to Si wafer surface[40]. For fabrication of Si nanostructures, a chemical resist (SEP 520 A, Zeon Corp.) was first spin-coated on the Si layer and a structural pattern was drawn by electron beam lithography (ELS-7700T Elionix Inc.). Next, a Cr mask with a thickness of 30 nm was evaporated on the sample by an electron beam evaporator, and Cr mask patterns were formed after lifting off resist. Then, the Si layer was selectively etched by using SF$_6$ and C$_4$F$_8$ plasma gases in a reactive ion etching chamber while applying a bias voltage between the plasma source and the target sample. Finally, the Cr mask was removed by immersing the sample in a di-ammonium cerium(IV) nitrate solution.

**Optical setup (see Supplementary Fig. 1).** A commercial laser-scanning microscope (FV-300, Olympus, Japan) was combined with a dark-field inverted microscope (IX-71, Olympus, Japan) to detect various kinds of scattering signals,

including dark-field image, scattering spectrum, scattering nonlinearity, Raman scattering, and oil-immersion backward scattering. Their corresponding setups are explained below:

*Dark-field image observation (Fig. 1c):* a broadband halogen lamp source was adopted with a dark-field high-NA condenser (U-UCD8, NA = 0.9, Olympus, Japan), a low-NA objective (UplanApo ×10, NA = 0.4, Olympus, Japan), and an eyepiece camera.

*Scattering spectrum acquisition (Supplementary Fig. 3):* with the same lamp, condenser, and objective, the forward scattered light was guided through FV-300, whose galvo mirrors selected scattering from a specific nanoblock, and then collected by a monochromator (Acton SpectraPro 300, Princeton, NJ) equipped with a cooled camera detector (iDus401, Andor, UK).

*Dark-field laser scanning to characterize point spread function and single-nanoblock nonlinearity (Fig. 1d and Fig. 2):* the excitation source was a 561-nm, 150-mW laser (Jive, Cobolt, Sweden). The same objective, condenser, and scanning unit were used. The scanning speed was 4 μs/pixel. The forward scattered light was collected by the dark-field condenser, and delivered by a built-in transmission photomultiplier tube connected with a liquid light waveguide.

*Oil-immersion nanoblocks with laser scanning (Supplementary Fig. 12):* backscattering was collected through reflection confocal paths, i.e., descanned by galvo mirrors, reflected by a beamsplitter, confocal filtered by a pinhole, and detected by a PMT in the reflection path. Different from the sample in air, where backscattering interferes with quartz/air interface reflection, here the immersion oil met the refractive index of the quartz substrate, resulting in background-free detection.

*Raman scattering measurement (Fig. 3a):* The same single-frequency 561-nm laser was used, and Raman scattering was epi-detected with the same ×10 objective and descanned with the same FV-300 scanning unit toward the same spectrometer. In front of the spectrometer, a notch filter and a slit were placed to block the residual reflected laser.

**Refractive index of silicon at elevated temperature.** The temperature-dependent complex refractive index of Si was experimentally measured up to 700 K by a commercial ellipsometer equipped with a sample heater (M-2000 DI ellipsometer, JA Woollam, NE). The sample for the ellipsometry experiment was a 150-nm silicon thin film on quartz, with width and length both equal to 1 cm. When deriving index from ellipsometric reflection measurement, the parameters of film thickness = 148 nm and surface roughness = 3 nm delivered the best fitting results.

To extrapolate the complex refractive index toward 1000 K, as shown in Supplementary Fig. 9. the following equations were adopted[41],

$$n(T) = a_n(T - 300) + n_n. \tag{1}$$

$$k(T) = a_k \exp\left(\frac{T}{T_k}\right) \tag{2}$$

where $n(T)$ and $k(T)$ were the real and imaginary parts of the refractive index. By fitting the experimentally obtained value with the formula, the constants $a_n$, $n_n$, $a_k$, $T_k$ were determined as $2.88 \times 10^{-4}$, 3.98, $1.04 \times 10^{-2}$, 353[K], respectively, all in good agreement with published results[41]. Using these constants, the complex refractive index from 300 to 1000 K was used for the scattering calculations.

**Simulation method.** For three-dimensional scattering calculations, we performed finite-difference time-domain simulations (Lumerical, Inc., FDTD Solutions). A Si nanostructure with a height of 150 nm, and variable widths of $w_x$ and $w_y$ was placed on a quartz substrate (the refractive index was fixed at 1.46), and each boundary in the total calculation domain was surrounded by perfect matching layers. A pulsed plane electromagnetic wave defined by the total-field scattered-field source was irradiated from the air side ($n_{air} = 1$) to the quartz substrate side with the surface normal. The forward scattering cross section $C_{scaF}$ and the absorption cross section $C_{abs}$ were detected by monitor layers surrounding the Si nanostructure. The monitor layer for the forward scattering was set at a distance of 500 nm and a width of 700 nm from the center of a Si nanostructure, corresponding to the numerical aperture of condenser in experiment.

The separation of scattering contribution from dipoles and quadrupoles was carried out by using multipole decomposition analysis (Supplementary Fig. 2). For the analysis, we calculated total and scattering fields in a Si nanostructure by using finite element method simulations (COMSOL, Inc., COMSOL Multiphysics). The substrate was ignored for simplicity in the calculations.

**Calculation of laser-induced temperature rise and relaxation.** For thermal calculations in the CW illumination (Fig. 3), the laser heat flux F of a Gaussian profile was defined in finite element simulations as,

$$F = \frac{2P_{in}}{\pi w_b^2} \exp\left(\frac{-2\left((x - x_0)^2 + (y - y_0)^2\right)}{w_b^2}\right) \tag{3}$$

where $w_b$ is a beam intensity radius at which the intensity is $\frac{1}{e^2}$, $x_0$ and $y_0$ are the center position of the beam, and $P_{in}$ corresponds to the laser power. The heat rate

**Table 1 Material properties for thermal simulation.**

|         | $\rho$ (kg·m$^{-3}$) | $C_p$ (J·kg$^{-1}$·K$^{-1}$) | $k$ (W·m$^{-1}$·K$^{-1}$) (as a function of T) |
|---------|----------------------|------------------------------|-----------------------------------------------|
| Silicon | 2329                 | 712                          | 260–20                                        |
| Quartz  | 2648                 | 739                          | 1.29–11.52                                    |
| Air     | 1.161                | 1000                         | 0.026–0.094                                   |
| Oil     | 1050                 | 1760                         | 0.14–0.016                                    |

$Q$ is calculated by $(F \cdot C_{abs})/V$, where $C_{abs}$ is the absorption cross section and $V$ is the volume of a Si nanostructure. The temperature $T$ is then calculated by Fourier's heat equation,

$$\rho C_p \frac{\partial T}{\partial t} + k\nabla^2 T = Q \qquad (4)$$

where $\rho$ is density, $C_p$ is the specific heat capacity at constant pressure, and $k$ is thermal conductivity. When considering CW illumination that the temperature reaches a steady state, the equation converges into Poisson's equation $k\nabla^2 T = Q$.

The parameters used in the simulation are in Table 1[42–46]. The heat capacity and the density of silicon as well as of the surrounding media were set as constants at 300 K. The temperature-dependent thermal conductivities were obtained from experimental literature. The temperature outside the calculation domain was fixed at 300 K, and the initial temperature in the calculation domain was 300 K.

During laser irradiation, temperature of silicon and the surrounding medium increased. Since $C_{abs}$ was dependent on the temperature, the final temperature of a Si nanostructure could not be determined at one calculation step, thus iterative calculation was necessary. We started by giving the heat rate $Q$ to a Si nanostructure, and using absorption cross section $C_{abs}(T_0)$ at an initial temperature $T_0 = 300$ K. The subsequent heating process was cut into discrete temperature steps $T_1, T_2, \ldots, T_i$. Based on the ellipsometry data and the method mentioned in the last section, new absorption cross section $C_{abs}$ at each temperature was sequentially calculated, until $T_i - T_{i+1}$, which indicated that the steady-state temperature inside a Si nanostructure was reached during laser irradiation. In all calculations, we confirmed that $|(T_{20} - T_{19})/T_{20}| \ll 1.0\%$. By this iteration approach, we could calculate the steady-state temperature and corresponding scattering cross section for each nanoblock size and excitation intensity.

The thermal relaxation in the Si nanostructure (Fig. 4b) was evaluated by the time-dependent thermal equation $\rho C_p \frac{\partial T}{\partial t} + k\nabla^2 T = Q$, in which the heat rate $Q$ was set to zero because the strong femtosecond pump pulse induces transient heating. The initial temperatures of the Si nanostructure and the surrounding domains were set to 1500 and 300 K, respectively. Since the thermal conductivity of quartz is much larger than oil, the relaxation is dominated by quartz. Based on the temperature-dependent index of silicon at $\lambda = 785$ nm[47], and by solving the time-dependent equation numerically, we derived the temperature relaxation averaged in a single nanoblock.

**Temperature measurement using Raman spectrum.** It is well known that when the temperature of crystallized silicon elevates, Raman spectra exhibit peaks shift, linewidth expansion, and anti-Stokes/Stokes ratio decrease[48]. Here we use the last parameter to determine temperature due to the inadequate resolution of our spectrometer. From the reference, an exponential dependence exists between the ratio of anti-Stoke over Stoke peak intensity and temperature, as given here

$$I_A/I_S = \exp\left(-\frac{\hbar\omega_0}{kT}\right) \qquad (5)$$

where $I_A$ and $I_S$ are intensities of Anti-Stoke and Stoke respectively, $\hbar$ is reduced Planck constant, $\omega_0$ is optical phonon frequency of silicon, $k$ is Boltzmann constant and $T$ is temperature. From our experiment in Fig. 3a, the ratio of $I_A/I_S$ increases from 0.23 at 1.3 mW μm$^{-2}$, 0.43 at 3.7 mW μm$^{-2}$, to 0.57 at 6.0 mW μm$^{-2}$. Therefore, the three numbers suggest a two-stage temperature increase of 610 and 380 K, respectively, manifesting dramatic photothermal effect.

**Transient measurements using ultrafast techniques.** A single Si nanoblock with a side width of 290 nm, immersed in oil, was selected to satisfy the resonant condition of the 785 nm laser (Supplementary Fig. 12). The transient backscattering of the silicon nanoblock was determined with pump-probe measurements by using a Ti:sapphire oscillator. The repetition rate was reduced to 8 MHz by using a pulse picker. The central wavelength was 785 nm with the full-width-at-half-maximum of ~12 nm. A long-wave-pass filter (LP02-785RE-25, Semrock) and a short-wave-pass filter (SP01-785RU-25, Semrock) at 785 nm were used to reshape the spectra of the optical pulses from the oscillator for pump and probe, respectively. The central wavelengths of the pump and probe were ~789 and ~772 nm, respectively. The collinear pump/probe beams were focused onto the samples with an oil-immersion objective lens (NA = 1.4). The sample, a Si nanoblock with a side width of 290 nm and thickness of 150 nm, was also immersed in oil to minimize reflection from the quartz substrate. The backscattered probe beams were collected by a beamsplitter, and then focused into a pinhole in front of a PMT. A short-wave-pass filter before the PMT was used to block the residual pump beam. An electro-optical

modulator was used to modulate the pump beam at 100 kHz. The signals, demodulated from the PMT with a lock-in amplifier, were recorded as a function of pump-probe delay. The detailed experimental setup is referred to our recent publication[49]. The duration of the pump/probe pulses at the samples was ~1 ps. The diameter of the spot size, defined by a Gaussian beam with $1/e^2$ in intensity, was ~510 nm. The pump fluence was 55 mJ cm$^{-2}$ while the probe fluence kept at ~ 1 mJ cm$^{-2}$. Dependence of repetition rate between 8 MHz and 200 kHz was conducted to confirm the coincidence of the transient responses.

When comparing our results using CW lasers to previous pump-probe results that used pulsed lasers, we recalculated their intensities as:

$$\text{Intensity} = \frac{\text{pulse energy}}{(\text{pulse width})(\text{focus area})}$$
$$= \frac{\text{laser power}}{(\text{repetition rate})(\text{pulse width})(\text{facus area})} \qquad (6)$$

**Saturated excitation microscopy (SAX) to enhance resolution.** The setup used in this experiment is a laser-scanning inverted microscope in Supplementary Fig. 1 plus home-built modulation and detection units. An explicit setup is shown in Supplementary Fig. 13. The light source was a continuous-wave laser at 532 nm. The modulation unit was composed of two acoustic optical modulators (AOMs) (AOM-402-AF3, IntraAction, IL), whose first-order diffraction beams were interfered to generate a temporally pure sinusoidal modulation at 10 kHz. The modulated beam was then raster scanned by a pair of galvanometer-mirrors and focused on the sample by an objective lens (UPlanSApo ×100/1.40 Oil, Olympus, Japan). Another beamsplitter in the reflection path directs scattered light from silicon nanoblock to the photomultiplier tube (PMT, H7710-13, Hamamatsu, Japan) through a confocal detection system.

For the detection unit, if the scattering signal was saturated, it carried not only the fundamental modulation frequency $f = 10$ kHz but also high-order harmonics 2$f$, 3$f$, etc. The electric output of the PMT was then fed to a lock-in amplifier (HF2LI, Zurich Instruments, Switzerland) to filter out the harmonic components. A photodetector that records the fundamental modulation frequency provides a reference to the lock-in amplifier. A computer system then processes individual harmonic signals to form two-dimensional scanning images in synchronization with the scanner.

## Data availability
The data that support the plots within this paper, the home-built codes that generate the simulated results, and other findings of this study are available from the corresponding author upon reasonable request.

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

## Acknowledgements

This work was supported by the Outstanding Young Scholarship Project of Ministry of Science and Technology, Taiwan, under grant MOST-105-2628-M-002-010-MY4 (SWC), 108-2321-B-002-058-MY2 (SWC), 109-2112-M-002 -026 -MY3 (SWC), and 108-2112-M-001-027 (KHL), as well as supported by the Higher Education Sprout Project funded by the Ministry of Science and Technology and Ministry of Education in Taiwan. S.W.C. acknowledges the generous award from the Foundation for the Advancement Outstanding Scholarship. This work was also supported by the Photonics Center at Osaka University and Japan Society for the Promotion of Science (JSPS) Core-to-Core Program, A. Advanced Research Networks. Y.N. was supported by a JSPS Research Fellowship for Young Scientists. A part of this work was supported by the "Nanotechnology Platform Project (Nanotechnology Open Facilities in Osaka University)" of the Ministry of Education, Culture, Sports, Science and Technology, Japan [No. F-17-OS-0011 and S-17-OS-0011]. We would like to thank Dr Yu-Ming Chang for his helpful discussions on Raman scattering, Mr. Guan-Jie Huang and Mr. Yu-Feng Chien for their helpful assistance in experiments. The ellipsometry measurement was kindly supported by Prof. Hsiang-Lin Liu and Mr. Hsiao-Wen Chen. We also would like to thank Shin-Etsu Chemical Co., Ltd. for donating high-quality silicon-on-quartz substrates.

## Author contributions

Y.S.D. and H.X.D. designed and performed most of the experiments. Y.N. was responsible for simulations. Samples were prepared by Y.N., H.I., J.H.Y., Y.P.L., K.P.C., and J.T. Y.L.T. and P.H.W. helped data analysis. K.F. assisted in Raman measurement and image acquisition. H.Y.C. and K.H.L. designed and performed the transient measurements. Y.N. and Y.S.D. wrote the initial draft of the manuscript. C.W.C. and S.W.C. revised it. All the authors discussed the results and contributed to the writing of the manuscript. J.T. and S.W.C. supervised the project.

## Competing interests

The authors declare no competing interests.
