## [Peer Review File · Nature Communications]

Reviewers' comments:

Reviewer #1 (Remarks to the Author):

Review of "Giant optical nonlinearity in single silicon nanostructure: ultrasmall all-optical switch and super-resolution"

The authors describe in their paper the strong impact of refractive index change by absorption heating on the scattering intensity of nanoscale Mie-resonators. Depending on the size of the resonators a substantial positive or negative deviation from the usual linear behaviour is observed which was clearly observed in dark field microscopic measurements. Evaluating the local temperature of the Mie-resonators from Stokes-Antistokes Raman spectra the authors could attribute this to the temperature dependent refractive index change of the Si. Taking also the temperature dependent absorption cross section of the Mie-resonators into account a simulation of the temperature increase in the Mie-resonators was carried out. The modelled scattering of the Mie-resonators based on this temperature affected refractive index then lead to a very good correspondence with the experimental results. Finally the fast switching of the nonlinearity could be demonstrated with time resolved pump probe measurements and a narrowing of the point spread function due to the linearity was also presented.

The experimental results and theoretical simulations are convincing and present a coherent picture of the effect. The fast relaxation of the nonlinearity in the ns- range due to the small overall thermal capacity (fast cool down) is surprising and demonstrates impressively that thermal switching can be fast, when it is applied to nanoscale volumina. In addition the results show, that high Q-resonances (e.g. from photonic crystals defects) are not necessary to employ nonlinearities and that small simple Mie-resonators can already be of use. With the current interest in Mie-resonator based dielectric metamaterials the topic of the paper appears timely. The added information in the supplementary material supports the claims of the paper further and demonstrates that the authors investigated the presented effects thoroughly and extensively.

Overall the paper reads well and should be accessible to a large community of readers.

However there are still a few shortcomings which the authors must address:

1) The critical point for the understanding that there is for some particles a negative nonlinear deviation of scattering intensity (100nm, 190nm) and for the 170nm a positive nonlinear deviation of scattering intensity is not given in the paper. In the moment it only appears in the supplementary material as caption on Fig E3: "Since Mie scattering spectrum red-shifts at elevated temperature, this explains...". This crucial explanation has to be also well presented in the main paper! Only then the reader can understand why the nonlinearity acts sometimes as increasing and sometimes as decreasing. The spectral shape of the scattering spectra with the different resonances is crucial here.

2) What is the influence of the free carriers on the refractive index. In the paper only the thermal impact on the refractive index is considered. However as the applied light wavelength (561nm) is well absorbed by the Si, a lot of free carriers are generated and should also have an impact on the refractive index or can this be neglected? The authors should estimate the impact of the free carrier concentration on the refractive index and describe the outcome of it in the supplementary material.

3) Fig 4c and explanation of resolution enhancement by SAX in the supplementary materials:
From the the explanation in the supplementary materials it is not clear how the resolution enhancement is exactly measured and how the curve in Fig. 4c of the main paper is actually obtained. It is mentioned that the two beams from the AOMs are interfered and "generate a sinusoidally modulated illumination beam at 10kHz frequency". Does this mean that a spatially modulated light pattern is generated whose intensity is additionally oscillating with a frequency of 10kHz in the time domain? The authors should explain in more detail how the enhanced resolution (reduced PSF) is actually measured since from the existing explanations this appears to be a quite complex procedure. Maybe a schematic drawing of the setup, which was used for the SAX-measurements will help.

If the authors can improve/clarify these mentioned points, I recommend the paper for publication.

Reviewer #2 (Remarks to the Author):

In the paper "Giant optical nonlinearity in single silicon nanostructure: ultrasmall all-optical switch and super-resolution imaging" authors show the all-optical tuning of the Si-blocks that support the excitation of Mie resonances. The paper is well written and explains everything in a detailed manner. It shows comprehensive studies on the photothermal nonlinearity in silicon nanoblocks, that originate from the Mie resonance excitation. Authors provide a large amount of data including nonlinear measurements for different dimensions of silicon blocks, Raman measurement, ellipsometry measurements, and others. Authors utilize Mie resonances to enhance the absorption of Silicon and to achieve 5-orders of magnitude enhancement of nonlinearity. Paper looks interesting for the metamaterial community, however, I am concerned about novelty and if it is suitable for Nature communications, as well as believe this paper requires some revisions.

1) As for the novelty please read recently published paper, Berzinš, Jonas, et al. "Laser-induced spatially-selective tailoring of high-index dielectric metasurfaces." *Optics Express* 28.2 (2020): 1539-1553.

Apart from the spectral tuning of the Mie resonances due to the heating, it also provides additional data about the melting/reshaping and damage threshold of Silicon metasurface. Please, emphasize the differences in your results.

2) Moreover, another paper has relevant studies that you do not cite. It provides the result of the interplay between free-carrier refractive index change, and thermal heating in Si-metasurface: Della Valle, Giuseppe, et al. "Nonlinear anisotropic dielectric metasurfaces for ultrafast nanophotonics." *ACS Photonics* 4.9 (2017): 2129-2136.

3) And another paper that can be cited: Bosch, M., et al. "Polarization states synthesizer based on a thermo-optic dielectric metasurface." *Journal of Applied Physics* 126.7 (2019): 073102.

Other questions:

1) The authors provide the main experimental results in the numbers of NDR – nonlinear deviation. I can see how it can have a mathematical meaning, but from the experimental point of view, this result can lead readers to confusion. For example, line 98, authors claim that you have "400% enhancement or 70% reduction of scattering, i.e.". Yes, it is defined that " i.e. deviation", but, it's not a 400%/70% change of scattering. I can accept NDR after I read what does it mean; however, if I think carefully, you calculate deviation from some number that cannot be achieved in the experiment. Moreover, the most important result is shown in Fig.3 – change of scattering efficiency of the nanoblock under different excitation power. Here we can clearly see that scattering changes and NRD has nothing in common, – the main changes of scattering occur at 190 nm width of the block. But NDR for 190 nm block has the smallest value. Therefore, please provide some other papers that use NDR value, since I admit, that I might not be familiar enough with that area, or please, consider emphasizing other experimental values.

2) Moreover, when you provide a comparison with other works, you use NDR. None of these papers in refs [19,21] use this value to describe the modulation of the signal. There are two values that majority of community uses: it is the absolute value modulation of optical signal (ΔI), or the relative modulation of the signal, $\Delta I/I_0$, where $\Delta I = I_{\text{pump}} - I_0$, I_0 is the unperturbed reflectance/transmittance and I_{pump} is the reflectance/transmittance when high-intensity laser is used – mostly for better visualization of the changes (Please see papers from references [19,21], or another paper that also shows I-scan, and authors use absolute values: Zubyyuk, Varvara V., et al. "Low-power absorption saturation in semiconductor metasurfaces." *ACS Photonics* 6.11 (2019): 2797-2806.) Therefore, please correct the comparison of the results with the previous works.

3) Since you use CW laser, and other papers use pulsed laser, please, provide how you recalculated their excitation intensity, for example, in supplementary materials.

4) I believe that the non-gaussian form of your results can come from the spectral tuning of the resonance. If you change the refractive index, the spectral position of the Mie resonances will tune, as I mentioned in the paper Berzinš, Jonas, et al. "Laser-induced spatially-selective tailoring of high-index dielectric metasurfaces." *Optics Express* 28.2 (2020): 1539-1553, or any other refractive index change papers. It will be relevant to show numerical calculation of spectrum at least for one sample, to show how it is spectrum is going to be modified when you change the refractive index of silicon due to the photothermal effect. Therefore, it will give a better explanation of the achieved results.

5) Please, provide more details regarding the nanoblock sample used:

a) How far are the nanoblocks from each other?

b) Is there any coupling between them that can modify the spectra of Mie resonances?

6) Other small corrections:

When you provide the number of nonlinearities, for example on line 64/66, please provide references.

I believe lines 88-90 are in general incorrect. In this review you separate metasurfaces from nanoresonators, but if I am not mistaken, in all this works authors use metasurfaces as arrays of nanoresonators.

Also, I recommend considering changing the title of the paper, since there is no "photothermal" anywhere.

Altogether, the work is of interest to the metamaterial/silicon community and can be published after revision.

Reviewer #3 (Remarks to the Author):

In their manuscript, Duh et al. report on the photothermal response of Mie-resonant silicon nanoparticles. I think this is a technically sound study revealing an interesting property of a well-studied system: light can efficiently 'tune' itself in an ultra-small particle by heating and exploiting the thermo-optic response of silicon. The authors proceed to demonstrate several intriguing applications of this effect for switching the scattering efficiency and reducing the spot size of a focused laser beam down to 132 nm at a wavelength of 592 nm. I think that the presentation quality, the rigor of the research effort, as well as the timeliness of the topic can secure publication in *Nature Communications* after the following issues are addressed:

1. The title of the work needs to be more specific about the nature of the nonlinearity. I suggest using the following wording: "Photothermal nonlinearity." Also, there is no need to emphasize the fact that it is an ultrasmall all-optical switch: the fact that a small silicon nanoparticle may show all-optical switching was discussed at length in Refs [19, 21, 26]. Finally, I am not convinced that the paper shows any explicit 'super-resolution imaging;' a mere tight focal spot demonstration in Fig. 4 is not enough to claim super-resolution imaging, especially in the title. Therefore, I recommend changing the title of the paper to "Giant photothermal nonlinearity in a single silicon nanoparticle." This way is it significantly more concise, and the main novelty is properly emphasized.

2. The way the silicon films were obtained is unclear. The authors used "a 150-nm-thick monocrystalline Si layer on the quartz substrate." Having a monocrystalline Si film on a quartz substrate is unusual. I could not locate this product on the website of the supplier the authors acknowledged. Therefore, I suggest adding to the Methods section of the paper the procedure of how these silicon films were deposited, for the sake of study reproducibility.

3. Please use caption to annotate the variables in Fig.1, such as C_{scaF} and C_{abs} .

4. Several stylistic suggestions would include getting rid of informal expressions, such as 'aka' (line 38), 'etc' (line 59), as well as unnecessary neologisms such as 'meta-silicon-material' (line 81), 'nano-silicon applications' (line 103) and others. In general, I find that the English of the paper is rough, and I strongly recommend invoking an English-editing service.

Reviewer #1 (Remarks to the Author):

The authors describe in their paper the strong impact of refractive index change by absorption heating on the scattering intensity of nanoscale Mie-resonators. Depending on the size of the resonators a substantial positive or negative deviation from the usual linear behaviour is observed which was clearly observed in dark field microscopic measurements. Evaluating the local temperature of the Mie-resonators from Stokes-Antistokes Raman spectra the authors could attribute this to the temperature dependent refractive index change of the Si. Taking also the temperature dependent absorption cross section of the Mie-resonators into account a simulation of the temperature increase in the Mie-resonators was carried out. The modelled scattering of the Mie-resonators based on this temperature affected refractive index then lead to a very good correspondence with the experimental results. Finally the fast switching of the nonlinearity could be demonstrated with time resolved pump probe measurements and a narrowing of the point spread function due to the linearity was also presented.

The experimental results and theoretical simulations are convincing and present a coherent picture of the effect. The fast relaxation of the nonlinearity in the ns- range due to the small overall thermal capacity (fast cool down) is surprising and demonstrates impressively that thermal switching can be fast, when it is applied to nanoscale volumina. In addition the results show, that high Q-resonances (e.g. from photonic crystals defects) are not necessary to employ nonlinearities and that small simple Mie-resonators can already be of use. With the current interest in Mie-resonator based dielectric metamaterials the topic of the paper appears timely. The added information in the supplementary material supports the claims of the paper further and demonstrates that the authors investigated the presented effects thoroughly and extensively.

Overall the paper reads well and should be accessible to a large community of readers. However there are still a few shortcomings which the authors must address:

1) The critical point for the understanding that there is for some particles a negative nonlinear deviation of scattering intensity (100nm, 190nm) and for the 170nm a positive nonlinear deviation of scattering intensity is not given in the paper. In the moment it only appears in the supplementary material as caption on Fig E3: "Since Mie scattering spectrum red-shifts at elevated temperature, this explains...". This crucial explanation has to be also well presented in the main paper! Only then the reader can understand why the nonlinearity acts sometimes as increasing and sometimes as decreasing. The spectral shape of the scattering spectra with the different resonances is crucial here.

We appreciate the reviewer's positive comments and we agree with the reviewer that spectral information of the nanoparticles at different temperatures should be given to enhance the clarity. In the revised figure 3e, 3f, and 3g, we have added temperature-dependent spectral shifts to explain the physical origin of the negative/positive nonlinear deviations.

Fig. 3 e-g...The inset of each figure presents the corresponding resonance spectrum shift at elevated temperatures.

(p. 9, third paragraph)

The negative nonlinear deviation (100-nm, 190-nm) and positive nonlinear deviation (170-nm) are well explained by the temperature-dependent resonance spectrum shift in the insets.

2) What is the influence of the free carriers on the refractive index. In the paper only the thermal impact on the refractive index is considered. However as the applied light wavelength (561nm) is well absorbed by the Si, a lot of free carriers are generated and should also have an impact on the refractive index or can this be neglected? The authors should estimate the impact of the free carrier concentration on the refractive index and describe the outcome of it in the supplementary material.

This is a good point. In pulsed laser excitation scenario (Fig. 4b), we have explicitly shown that two relaxations in time were observed. Free carriers are first excited by pulsed lasers, and they lead to index changes. Subsequently, the temperature rises due to the carrier-phonon scatterings and recombination of free carriers. The fast relaxation (10s of picoseconds) corresponds to free carriers and the subsequently slow relaxation (nanosecond) represents thermal effect. That is, free carriers do have an impact on refractive index, as the reviewer suggested, but the effect can be clearly distinguished from thermal response in time domain.

Furthermore, in Fig. 2 and 3 of the manuscript, we used CW lasers, whose intensity is 4-5 orders lower than the peak intensity of mode-locked laser excitation in Fig. 4b. Since the density of free carrier should be proportional to excitation intensity, in CW excitation case, free carrier effect should be negligible. This is verified by the outstanding agreement between experiment (Fig. 2a-2c) and photothermal theory (Fig. 3e-3g).

(p. 12, first paragraph)

... The response of the free carrier and photothermal effect is clearly distinguished in time domain. Note that in Fig. 2 and Fig. 3, where CW lasers are applied, the laser intensity is 4-5 orders lower than that of Fig. 4b, and thus, the free carrier effect is negligible....

3) Fig 4c and explanation of resolution enhancement by SAX in the supplementary materials: From the the explanation in the supplementary materials it is not clear how the resolution enhancement is exactly measured and how the curve in Fig. 4c of the main paper is actually

obtained. It is mentioned that the two beams from the AOMs are interfered and “generate a sinusoidally modulated illumination beam at 10kHz frequency”. Does this mean that a spatially modulated light pattern is generated whose intensity is additionally oscillating with a frequency of 10kHz in the time domain? The authors should explain in more detail how the enhanced resolution (reduced PSF) is actually measured since from the existing explanations this appears to be a quite complex procedure. Maybe a schematic drawing of the setup, which was used for the SAX-measurements will help.

We apologize for not explaining the setup clearly. The modulation is only in temporal domain, not in spatial domain, and the purpose of using two AOMs is simply to generate a pure sinusoidal modulation [see ref. 31 of the main text]. Following the reviewer’s suggestion, we have added a setup scheme in supplementary Fig. 13, to explain in detail how to achieve the enhanced resolution.

If the authors can improve/clarify these mentioned points, I recommend the paper for publication.

Reviewer #2 (Remarks to the Author):

In the paper "Giant optical nonlinearity in single silicon nanostructure: ultrasmall all-optical switch and super-resolution imaging" authors show the all-optical tuning of the Si-blocks that support the excitation of Mie resonances. The paper is well written and explains everything in a detailed manner. It shows comprehensive studies on the photothermal nonlinearity in silicon nanoblocks, that originate from the Mie resonance excitation. Authors provide a large amount of data including nonlinear measurements for different dimensions of silicon blocks, Raman measurement, ellipsometry measurements, and others. Authors utilize Mie resonances to enhance the absorption of Silicon and to achieve 5-orders of magnitude enhancement of nonlinearity. Paper looks interesting for the metamaterial community, however, I am concerned about novelty and if it is suitable for Nature communications, as well as believe this paper requires some revisions.

1) As for the novelty please read recently published paper, Berzinš, Jonas, et al. "Laser-induced spatially-selective tailoring of high-index dielectric metasurfaces." *Optics Express* 28.2 (2020): 1539-1553.

Apart from the spectral tuning of the Mie resonances due to the heating, it also provides additional data about the melting/reshaping and damage threshold of Silicon metasurface. Please, emphasize the differences in your results.

We appreciate the reviewer to compare our work with recently published papers. The main difference is that the OE paper describes a non-reversible effect, where the silicon nanoparticle shape and the corresponding Mie resonance are permanently changed by picosecond laser induced heat. In contrast, our photothermal nonlinear effects do not involve any nanostructure shape change, and are fully reversible.

(p. 6, last paragraph)

The reversibility and repeatability of the nonlinear responses are demonstrated in Fig. 2e and Fig. E6, thus excluding the possibility of **non-reversible oxidation or shape change** of silicon²⁴.

2) Moreover, another paper has relevant studies that you do not cite. It provides the result of the interplay between free-carrier refractive index change, and thermal heating in Si-metasurface: Della Valle, Giuseppe, et al. "Nonlinear anisotropic dielectric metasurfaces for ultrafast nanophotonics." *ACS Photonics* 4.9 (2017): 2129-2136.

We thank the reviewer to mention this reference, that we are aware of. Although this reference discussed photothermal effect of silicon nanoparticles, it aimed to minimize photothermal effect at selective wavelength to achieve high-speed modulation based on free carriers. In this reference, the modulation speed reaches picosecond scale, but modulation depth is less than 10% (by free carrier), at a much higher excitation intensity ($>GW/cm^2$, or equivalently $>10W/\mu m^2$). Their reported photothermally induced modulation is always less than 1%. In contrast, our work demonstrates $>400\%$ variation of scattering with a CW laser at only few $mW/\mu m^2$! That is, their nonlinear responses are much smaller than ours.

(p. 11, last paragraph)

Note that our reported photothermal nonlinear response in an isolated low-Q silicon resonator is much higher than that in a thermally coupled nanostructures on a metasurface **with free-carrier effect**³³ or with Fano resonance (Q-factor ~1000).¹⁹

3) And another paper that can be cited: Bosch, M., et al. "Polarization states synthesizer based on a thermo-optic dielectric metasurface." *Journal of Applied Physics* 126.7 (2019): 073102.

We have cited this article, along with several other thermo-optic papers, as a potential application direction in the future.

(p. 12, 3rd paragraph)

In addition to the above applications, recently, there has been an emerging trend using thermo-optic effect to modulate metamaterial optical properties, such as polarization³⁵, miniaturized image contrast³⁶, and metalens focal length³⁷. We envision that our result manifests the potential of pushing these various modulations toward all-optical control.

Other questions:

1) The authors provide the main experimental results in the numbers of NDR – nonlinear deviation. I can see how it can have a mathematical meaning, but from the experimental point of view, this result can lead readers to confusion. For example, line 98, authors claim that you have “400% enhancement or 70% reduction of scattering, i.e.”. Yes, it is defined that “i.e. deviation”, but, it’s not a 400%/70% change of scattering. I can accept NDR after I read what does it mean; however, if I think carefully, you calculate deviation from some number that cannot be achieved in the experiment. Moreover, the most important result is shown in Fig.3 – change of scattering efficiency of the nanoblock under different excitation power. Here we can clearly see that scattering changes and NDR has nothing in common, – the main changes of scattering occur at 190 nm width of the block. But NDR from 190 nm block has the smallest value. Therefore, please provide some other papers that use NDR value, since I admit, that I might not be familiar enough with that area, or please, consider emphasizing other experimental values.

We agree that no other literature use NDR, but we have a good reason for it. Before I explain our reason, I hope to clarify the meaning behind the reviewer’s statement: “(in Fig. 3) the main changes of scattering occur at 190 nm width of the block. But NDR from 190 nm block has the smallest value” Please note that the meaning of NDR is deviation from linear extrapolation, and it would be zero when there is no deviation. In Fig. 3, NDR from 190 nm block has the value of -70%, which represents a large deviation, in good agreement with the scattering change pattern in Fig. 3g. That is, NDR serves as a reasonable indicator of the nonlinearity that includes both amplitude and sign.

Conventionally, in the field of nonlinear optics, pulse lasers were adopted, and typically the pulse lasers are split into pump and probe beams. With pump-probe experiments, nonlinear response is defined as $\Delta I/I_0$, where I_0 is the signal intensity without pump pulse and ΔI is

that with pump pulse. In the case of measuring scattering, the pump-probe nonlinear response is defined as $\Delta S/S$.

Now in our case, we used a CW laser for efficient heating, and the same laser beam provides nonlinear characterization. Therefore, it is not necessary to split into pump and probe beams, and thus the definition of our nonlinear response would be different from most existing literatures. Having said this, as we explained in Fig. 2a, NDR is in fact the same as $\Delta S/S$, by defining S as linearly extrapolated value, and ΔS is deviation from S. To further emphasize this similarity, we have added the notion that $NDR = \Delta S/S$ in all relevant figures (Fig. 2a, 2b, 2c, 2f; Fig. 3d, 3e-g; Fig. 4b).

2) Moreover, when you provide a comparison with other works, you use NDR. None of these papers in refs [19,21] use this value to describe the modulation of the signal. There are two values that majority of community uses: it is the absolute value modulation of optical signal (ΔI), or the relative modulation of the signal, $\Delta I/I_0$, where $\Delta I = I_{\text{pump}} - I_0$, I_0 is the unperturbed reflectance/transmittance and I_{pump} is the reflectance/transmittance when high-intensity laser is used – mostly for better visualization of the changes (Please see papers from references [19,21], or another paper that also shows I-scan, and authors use absolute values: Zubyyuk, Varvara V., et al. "Low-power absorption saturation in semiconductor metasurfaces." ACS Photonics 6.11 (2019): 2797-2806.) Therefore, please correct the comparison of the results with the previous works.

As we mentioned in the last response, our definition of NDR is the same to $\Delta I/I_0$ in the previous references (I_0 is linearly extrapolated in our case). Therefore, we compare directly our NDR with the value of $\Delta I/I_0$ in other references.

3) Since you use CW laser, and other papers use pulsed laser, please, provide how you recalculated their excitation intensity, for example, in supplementary materials.

We have included the recalculation in the Methods section, under the title of "Transient measurements by ultrafast techniques" in page 16 of the main text, which is based on the following equation:

$$\begin{aligned} \text{Intensity} &= \frac{\text{pulse energy}}{(\text{pulse width})(\text{focus area})} \\ &= \frac{\text{laser power}}{(\text{repetition rate})(\text{pulse width})(\text{focus area})} \end{aligned}$$

For example, in Fig. 4 of Ref. 21, they achieved <1% transmission modulation by using pump fluence of 30 $\mu\text{J}/\text{cm}^2$, and 45 fs pulse width. The corresponding excitation intensity is $30/45 * 10^9 \text{ W}/\text{cm}^2 \sim 6.7 * 10^4 \text{ mW}/\mu\text{m}^2$, which is much larger than our excitation intensity in CW case.

4) I believe that the non-gaussian form of your results can come from the spectral tuning of the resonance. If you change the refractive index, the spectral position of the Mie resonances will tune, as I mentioned in the paper Berzinš, Jonas, et al. "Laser-induced spatially-selective tailoring of high-index dielectric metasurfaces." *Optics Express* 28.2 (2020): 1539-1553, or any other refractive index change papers. It will be relevant to show numerical calculation of spectrum at least for one sample, to show how its spectrum is going to be modified when you change the refractive index of silicon due to the photothermal effect. Therefore, it will give a better explanation of the achieved results.

We agree with the reviewer, and added simulated spectrum tuning in the revised Fig. 3e-3g, to provide a better explanation of the nonlinearity.

5) Please, provide more details regarding the nanoblock sample used:

- How far are the nanoblocks from each other?
- Is there any coupling between them that can modify the spectra of Mie resonances?

The distance between each nanoblock is 5 micrometers, and thus the coupling among them is negligible. The information is added in p. 3, last paragraph.

6) Other small corrections:

When you provide the number of nonlinearities, for example on line 64/66, please provide references.

We have added references for Kerr nonlinearity (ref 3) and photothermal nonlinearity (ref 5) of bulk silicon.

- Leuthold, J., Koos, C. & Freude, W. Nonlinear silicon photonics. *Nat. Photonics* 4, 535–544 (2010).
- Horvath, C., Bachman, D., Indoe, R. & Van, V. Photothermal nonlinearity and optical bistability in a graphene–silicon waveguide resonator. *Opt. Lett.* 38, 5036 (2013).

I believe lines 88-90 are in general incorrect. In this review you separate metasurfaces from nanoresonators, but if I am not mistaken, in all this works authors use metasurfaces as arrays of nanoresonators.

We thank the reviewer for careful reading. The sentence is modified as " For example, Si metasurfaces can be applied for fabrication of ultracompact phase controllers,¹⁹

several order-of-magnitude enhancement in third harmonic generation,^{20,21} and two-photon absorption.²²

Also, I recommend considering changing the title of the paper, since there is no “photothermal” anywhere.

Agree, we modify the title to be "Giant **photothermal** nonlinearity in a single silicon nanostructure", as suggested by the third reviewer

Altogether, the work is of interest to the metamaterial/silicon community and can be published after revision.

Reviewer #3 (Remarks to the Author):

In their manuscript, Duh et al. report on the photothermal response of Mie-resonant silicon nanoparticles. I think this is a technically sound study revealing an interesting property of a well-studied system: light can efficiently 'tune' itself in an ultra-small particle by heating and exploiting the thermo-optic response of silicon. The authors proceed to demonstrate several intriguing applications of this effect for switching the scattering efficiency and reducing the spot size of a focused laser beam down to 132 nm at a wavelength of 592 nm. I think that the presentation quality, the rigor of the research effort, as well as the timeliness of the topic can secure publication in Nature Communications after the following issues are addressed:

1. The title of the work needs to be more specific about the nature of the nonlinearity. I suggest using the following wording: "Photothermal nonlinearity." Also, there is no need to emphasize the fact that it is an ultrasmall all-optical switch: the fact that a small silicon nanoparticle may show all-optical switching was discussed at length in Refs [19, 21, 26]. Finally, I am not convinced that the paper shows any explicit 'super-resolution imaging;' a mere tight focal spot demonstration in Fig. 4 is not enough to claim super-resolution imaging, especially in the title. Therefore, I recommend changing the title of the paper to "Giant photothermal nonlinearity in a single silicon nanoparticle." This way is it significantly more concise, and the main novelty is properly emphasized.

Agree, we modify the title to be "Giant photothermal nonlinearity in a single silicon nanostructure", as suggested by the third reviewer

2. The way the silicon films were obtained is unclear. The authors used "a 150-nm-thick monocrystalline Si layer on the quartz substrate." Having a monocrystalline Si film on a quartz substrate is unusual. I could not locate this product on the website of the supplier the authors acknowledged. Therefore, I suggest adding to the Methods section of the paper the procedure of how these silicon films were deposited, for the sake of study reproducibility.

We have added a reference and description in the Methods section.
(p.12, last paragraph)

The 150-nm-thick monocrystalline silicon on quartz substrate (Shin-Etsu Chemical Co., Ltd.) was fabricated by wafer bonding at the temperature of $T < 1000$ degree after H⁺ ion implantation to Si wafer surface.³⁸

3. Please use caption to annotate the variables in Fig.1, such as CscF and Cabs.

The corresponding annotation is added in the caption of Fig. 1.

e. Simulated single-particle forward scattering (CscF) and f. absorption cross section (Cabs) spectra versus different nanoblock widths.

4. Several stylistic suggestions would include getting rid of informal expressions, such as 'aka' (line 38), 'etc' (line 59), as well as unnecessary neologisms such as 'meta-silicon-material' (line 81), 'nano-silicon applications' (line 103) and others. In general, I find that the English of the paper is rough, and I strongly recommend invoking an English-editing service.

The manuscript was English edited as requested.

REVIEWER COMMENTS

Reviewer #1 (Remarks to the Author):

The authors considered the remarks in the reviews and improved the manuscript accordingly. I therefore recommend to publish the paper now in its new form.

Reviewer #2 (Remarks to the Author):

Dear authors,

Thank you for your response. Even though the paper has a lot of exciting results, I believe they should be modified to be understood by the audience. Unfortunately, there are quite a lot of questions that arise when I read the paper. I will focus on two significant issues.

1) One of the main problems is NDR. As you mentioned, it is something that no one uses. "We agree that no other literature use NDR, but we have a good reason for it." And that fact creates large misunderstanding of the whole results of the paper and I believe that it isn't something that can be stated as the main result of the paper in the abstract and conclusions.

The reason for that is that apples should be compared to apples. And the use of NDR confused not only readers but authors too.

Despite all that is said, NDR is definitely not something that you can compare to pump/probe experiments. For example, paper "Shcherbakov, Maxim R., et al. "Ultrafast all-optical switching with magnetic resonances in nonlinear dielectric nanostructures." Nano letters 15.10 (2015): 6985-6990." provide not only pump-probe measurements with 1% change but also I-scan measurements for silicon metasurface. Here authors also mention thermal effects, similar to what you show in Fig. 2, I include the figure from the paper below, please see additional file.

Therefore, for that figure, we can also estimate NRD, and it will be at least 10% for top and bottom cases. Please note that the sign here is opposite because they measured here transmission. Therefore, the fact that NDR is something that no-one uses confused not only readers but also authors. Thus, the statement from the response letter "NDR is, in fact, the same as $\Delta S/S$, by defining S as linearly extrapolated value, and ΔS is a deviation from S . To further emphasize this similarity" is incorrect, same as a claim in the paper "74 However, the required intensity is $\sim 104 \text{ mW}/\mu\text{m}^2$, and the corresponding nonlinear deviation is only less than 1%". In my opinion, all shown results should be compared not to the pump-probe measurements but only to I-scan measurements.

2) The following question is related the comparison of the shown results with the previous works and power used.

As I mentioned before, for me, it's questionable if static results can be compared to pump-probe results. They can be if we compare values of the absolute change of reflectance/transmittance as it is done in all pump/probe works. No one compares $\Delta I/I$ between papers because it can be enormous when you divide by small value (dip in scattering, exactly what you have for some cases), and it has no meaning if we want to think about applications. $\Delta I/I$ is mostly used to visualize the changes more clearly. I strongly recommend to the authors to get rid of the NDR value, your results are excellent without it. Also, I agree that for nonlinear optic processes for pulsed lasers, we need to compare peak power with average power for the CW case. However, for thermal processes that have nanosecond-microsecond lifetime time to achieve the same amount of heating, we need to have the same average power. Please correct me if I am wrong.

3) Same as another reviewer, I am completely confused about the results from pump-probe measurements

in Fig.4. There are absolutely no details in supplementary, and actually, the reader should be referred to the Methods section. To claim that there is a heating process with ns switching time, all equations that you used for the model need to be shown, parameters of the simulations, etc. Or you can refer the reader to another paper, but at least you should write how theoretical data was achieved. Otherwise, none of these results can be reproduced. The main question is, how can you separate heating from carrier relaxation processes that also can be quite long (Augier recombination, for example). You mention that it is possible to distinguish the FC effect from thermal, but for the reader, it is impossible.

Moreover, the statement that claims that you are first to show photothermal relaxation with ns switching times is wrong.

320 "Here, we show for the first time, theoretically and experimentally, that photothermal relaxation time of an isolated silicon nanostructure reaches nanoseconds, i.e. GHz operation potential, with very large modulation depth, which has not been previously reported. "

Please see the paper "Della Valle G, Hopkins B, Ganzer L, Stoll T, Rahmani M, Longhi S, Kivshar YS, De Angelis C, Neshev DN, Cerullo G. Nonlinear anisotropic dielectric metasurfaces for ultrafast nanophotonics. ACS Photonics. 2017 Sep 20;4(9):2129-36."

In the abstract, authors mention that they have nanosecond lattice dynamics: "We reveal a set of operating windows where ultrafast all-optical modulation of transmission is achieved with full return to zero in 20 ps. This is made possible because of the distinct dispersive features exhibited by the competing nonlinear processes in transmission and despite the slow (nanosecond) internal lattice dynamics. "

4) Small details:

In Methods: thank you for providing a lot of details; however, section "Temperature estimation during laser irradiation" is about temperature, used rate equations that need to be written; otherwise it's all just describing.

Reviewer #3 (Remarks to the Author):

I have carefully studied the responses of the authors and re-read their revised manuscript. I think that the authors did a great job addressing all the issues raised by the reviewers, and the manuscript can be accepted for publication in its present form.

We would like to thank reviewer 1 and reviewer 3 for recommending publication of this work. Reviewer 2 also considered our work “has a lot of exciting results”, but suggested a few critical points for revision. In the following we provide a point-to-point response to reviewer 2’s suggestions.

1) One of the main problems is NDR. As you mentioned, it is something that no one uses. “We agree that no other literature use NDR, but we have a good reason for it.” And that fact creates large misunderstanding of the whole results of the paper and I believe that it isn’t something that can be stated as the main result of the paper in the abstract and conclusions.

The reason for that is that apples should be compared to apples. And the use of NDR confused not only readers but authors too. Despite all that is said, NDR is definitely not something that you can compare to pump/probe experiments. For example, paper “*Shcherbakov, Maxim R., et al. "Ultrafast all-optical switching with magnetic resonances in nonlinear dielectric nanostructures." Nano letters 15.10 (2015): 6985-6990.*” provide not only pump-probe measurements with 1% change but also I-scan measurements for silicon metasurface. Here authors also mention thermal effects, similar to what you show in Fig. 2, I include the figure from the paper below, please see additional file.

Therefore, for that figure, we can also estimate NRD, and it will be at least 10% for top and bottom cases. Please note that the sign here is opposite because they measured here transmission. Therefore, the fact that NDR is something that no-one uses confused not only readers but also authors. Thus, the statement from the response letter “NDR is, in fact, the same as $\Delta S/S$, by defining S as linearly extrapolated value, and ΔS is a deviation from S. To further emphasize this similarity” is incorrect, same as a claim in the paper “74 However, the required 85 intensity is $\sim 104 \text{ mW}/\mu\text{m}^2$, and the corresponding nonlinear deviation is only less than 1%”. In my opinion, all shown results should be compared not to the pump-probe measurements but only to I-scan measurements.

We understand that the reviewer emphasized the comparison between CW I-scan and femtosecond pump-probe experiment with the same $\text{NDR} = \Delta S/S$ definition may not be appropriate. One major difference is that NDR of the CW I-scan experiment is based on an extrapolated S (defined as S_e in the revised manuscript), while $\Delta S/S$ for femtosecond pump-probe is based on a really achievable S. Furthermore, in CW experiment, the temperature is at steady state, while in the femtosecond pump-probe case, thermal response is transient. To avoid misunderstanding among readers, and considering that $\Delta I/I$ is a typical indicator in the pump-probe field, we decide to define $\text{NDR} = \Delta S/S_e$ as a special indicator for the CW case, where S_e is the extrapolated linear response, and using $\Delta S/S$ for femtosecond pump-probe experiments. With this modification, this should avoid possible confusing to the readers for direct comparison of CW results to femtosecond pump-probe results.

We have removed the sentence in p. 4 “the required intensity is $\sim 10^4$ mW/ μm^2 , and the corresponding nonlinear deviation is usually less than 1%.” The definition of NDR is revised in the first paragraph of p. 6 as “We define the nonlinear deviation ratio (NDR) as $\Delta S/S_e$, where ΔS is the percentage deviation of measured scattering, and S_e is the extrapolated linear response.” All NDR marks in the corresponding figures are revised as well.

However, this does not mean $\Delta I/I$ in I-scan cannot predict the case of pump-probe. Reviewer 2 used the Nano Lett. paper as an example to state “all shown results should be compared not to the pump-probe measurements but only to I-scan measurements” based on the observation that in the paper the pump-probe results showed $\Delta T/T=1\%$ (Fig. 4 as cited below) but I-scan shows at least 10% NDR (Fig. 3a as cited below). We would like to note that the pump-probe measurements in the paper were conducted at low pump powers with fluence of 0.03 mJ/cm², so the difference results from different experimental conditions. As shown in Fig. 3a of the paper (attached below), $\Delta T/T$ is $\sim 10\%$ at 0.3 mJ/cm². Considering the contribution of TPA shows linear dependence at low pump fluence, $\Delta T/T$ should be $\sim 1\%$ at 0.03 mJ/cm². Therefore, by calibrating the excitation fluence, femtosecond I-scan shows comparable results to the femtosecond pump-probe results.

[redacted]

Regarding CW case, we would like to clarify that $\text{NDR} = \Delta S/S_e$ in the CW I-scan also predicts modulation depth $\Delta S/S$ in the CW pump-probe experiment. Fig. 4a in our manuscript shows that modulation depth $\Delta S/S$ reaches 90% in the CW two-color pump-probe experiment, where one CW beam (pump, 592-nm) modulates the other CW beam (probe, 543-nm, which determines the real S). This value is similar to the

$NDR = \Delta S/S_e$ that we observed in the CW I-scan (Fig. 2), which has virtual S_e and a single CW beam.

2) The following question is related the comparison of the shown results with the previous works and power used.

As I mentioned before, for me, it's questionable If static results can be compared to pump-probe results. They can be if we compare values of the absolute change of reflectance/transmittance as it is done in all pump/probe works. No one compares $\Delta I/I$ between papers because it can be enormous when you divide by small value (dip in scattering, exactly what you have for some cases), and it has no meaning if we want to think about applications. $\Delta I/I$ is mostly used to visualize the changes more clearly. I strongly recommend to the authors to get rid of the NDR value, your results are excellent without it. If not completely, then at least in the abstract.

We agree that it may not be appropriate to compare static results by using CW lasers and transient results by using femtosecond lasers. Therefore, in the main text, we have removed corresponding comparisons. As we mentioned when responding the first point, we feel that by defining NDR specific to CW I-scan experiment could avoid direct comparison between CW experiment and femtosecond pump-probe experiment in our manuscript. As the reviewer mentioned “ $\Delta I/I$ is mostly used to visualize the changes more clearly”. Therefore, we keep the NDR definition only for CW experiments, and using $\Delta S/S$ for femtosecond pump-probe experiments.

To remove NDR in the abstract, the corresponding sentence has been modified as “Here we show that Si nano-resonators exhibit a large photothermal nonlinearity, yielding 90% reversible and repeatable modulation from linear scattering response at low excitation intensities.”

Nevertheless, we cannot agree with the reviewer's argument that “No one compares $\Delta I/I$ between papers because it can be enormous when you divide by small value (dip in scattering, exactly what you have for some cases), and it has no meaning if we want to think about applications.”. $\Delta I/I$ reflects the percentage of changes in I. It doesn't matter if I is small or large. In our case, it directly reflects the percentage changes of the scattering cross-sections, which is meaningful.

Also, I agree that for nonlinear optic processes for pulsed lasers, we need to compare peak power with average power for the CW case. However, for thermal processes that have nanosecond- microsecond lifetime time to achieve the same amount of heating, we need to have the same average power. Please correct me if I am wrong.

Average power of pulsed lasers can only be compared to that of CW lasers if the thermal response is much slower than the pulse repetition rate. In our case, we have shown that the thermal relaxation in silicon nanoblocks is ~ 1 ns, which is much faster than the time interval of two pulses (>125 ns in our experimental condition). That is, temperature increase is from single pulse heating and the temperature recovers to room temperature before the next pulse. In contrast, CW laser heating is from an accumulation effect to reach an equilibrium temperature. Therefore, we do not compare the average power of femtosecond lasers and CW lasers because their heating mechanisms are different.

3) Same as another reviewer, I am completely confused about the results from pump-probe measurements in Fig.4. There are absolutely no details in supplementary, and actually, the reader should be referred to the Methods section. We have corrected the sentence to refer the readers to the Methods section.

To claim that there is a heating process with ns switching time, all equations that you used for the model need to be shown, parameters of the simulations, etc. Or you can refer the reader to another paper, but at least you should write how theoretical data was achieved. Otherwise, none of these results can be reproduced.

We have included equations and parameters for nanosecond thermal relaxation in the revised section of “**Calculation of laser-induced temperature rise and relaxation**” (p.15-16).

The main question is, how can you separate heating from carrier relaxation processes that also can be quite long (Auger recombination, for example). You mention that it is possible to distinguish the FC effect from thermal, but for the reader, it is impossible.

We agree that the transient signals due to the free carrier and lattice temperature cannot be fully distinguished especially in the regime of short time delay, say before 100 ps. Please note that the Auger recombination rate increases with increasing carrier density n as $dn/dt = Cn^3$, where C is Auger coefficient [REF 34]. Under high carrier concentration, the lifetime of Auger recombination in Si asymptotically reaches 6 ps [REF 34]. Additionally, we quote a figure and a sentence from [REF 35] “*The carriers relax by rapid Auger recombination and phonon emission, causing the lattice to heat in about 7 ps; the higher the fluence, the higher the temperature of the lattice (Fig. 4).*”

[redacted]

After the rapid recombination of most free carriers, the pump-probe signal is dominated by lattice temperature. Therefore, for nanosecond relaxation as we observed in Fig. 4b, it should be dominated by thermal response. We have rephrased the statement in p.12 of the main text to avoid misunderstanding

Looking into the experimental curve of Fig. 4b, there are apparently two relaxation processes. First, $\Delta S/S$ reaches -50% within the pulse duration (~ 1 ps) and subsequently relaxes to -35% within 0.1 ns. **Since the excited density of free carriers is high (10^{20} - 10^{21} cm $^{-3}$ under our experimental conditions) and close to the damage threshold, most energy of free carriers is efficiently transferred to the lattice temperature through the Shockley-Read-Hall process and Auger recombination in this duration.^{34,35,36} Following, $\Delta S/S$ is dominated by lattice temperature and reveals a second relaxation which takes a few nanoseconds to zero. This is primarily attributed to thermal dissipation from the Si nanoblock to the surrounding medium.**

34. Yoffa, E. J. Dynamics of dense laser-induced plasmas. *Phys. Rev. B* 21, 2415–2425 (1980).
35. Sundaram, S. K. & Mazur, E. Inducing and probing non-thermal transitions in semiconductors using femtosecond laser pulses. *Nat. Mater.* 1, 217–224 (2002).
36. Sabbah, A. J. & Riffe, D. M. Femtosecond pump-probe reflectivity study of silicon carrier dynamics. *Phys. Rev. B* 66, 165217 (2002).

Moreover, the statement that claims that you are first to show photothermal relaxation with ns switching times is wrong.

320 “Here, we show for the first time, theoretically and experimentally, that photothermal relaxation time of an isolated silicon nanostructure reaches nanoseconds, i.e. GHz operation potential, with very large modulation depth, which has not been previously reported. ”

Please see the paper “Della Valle G, Hopkins B, Ganzer L, Stoll T, Rahmani M, Longhi S, Kivshar YS, De Angelis C, Neshev DN, Cerullo G. Nonlinear anisotropic

dielectric metasurfaces for ultrafast nanophotonics. ACS Photonics. 2017 Sep 20;4(9):2129-36.”

In the abstract, authors mention that they have nanosecond lattice dynamics: “We reveal a set of operating windows where ultrafast all-optical modulation of transmission is achieved with full return to zero in 20 ps. This is made possible because of the distinct dispersive features exhibited by the competing nonlinear processes in transmission and despite the slow (nanosecond) internal lattice dynamics.”

We appreciate the careful comment from the reviewer and have removed the priority statement.

4) Small details:

In Methods: thank you for providing a lot of details; however, section “Temperature estimation during laser irradiation” is about temperature, used rate equations that need to be written; otherwise it’s all just describing.

Corresponding equations and parameters are added in the revised section (p. 15).

The temperature T is then calculated by Fourier’s heat equation,

$$\rho C_p \frac{\partial T}{\partial t} + k \nabla^2 T = Q$$

where ρ is density, k is thermal conductivity, C_p is the specific heat capacity at constant pressure. When considering CW illumination that the temperature reaches steady state, the equation converges into Poisson’s equation $k \nabla^2 T = Q$.

REVIEWERS' COMMENTS:

Reviewer #2 (Remarks to the Author):

Dear authors, thank you for your response. I recommend the paper for publication.